# Behavioral Homophily in Social Media via Inverse Reinforcement Learning: A Reddit Case Study

## Abstract

Online communities play a critical role in shaping societal discourse and influencing collective behavior in the real world. The tendency for people to connect with others who share similar characteristics and views, known as homophily, plays a key role in the formation of echo chambers which further amplify polarization and division. Existing works examining homophily in online communities traditionally infer it using content- or adjacency-based approaches, such as constructing explicit interaction networks or performing topic analysis. These methods fall short for platforms where interaction networks cannot be easily constructed and fail to capture the complex nature of user interactions across the platform. This work introduces a novel approach for quantifying user homophily. We first use an Inverse Reinforcement Learning (IRL) framework to infer users' policies, then use these policies as a measure of behavioral homophily. We apply our method to Reddit, conducting a case study across 5.9 million interactions over six years, demonstrating how this approach uncovers distinct behavioral patterns and user roles that vary across different communities. We further validate our behavioral homophily measure against traditional content-based homophily, offering a powerful method for analyzing social media dynamics and their broader societal implications. We find, among others, that users can behave very similarly (high behavioral homophily) when discussing entirely different topics like soccer vs e-sports (low topical homophily), and that there is an entire class of users on Reddit whose purpose seems to be to disagree with others.

**ACM Reference Format:**
Anonymous Author(s). 2024. Behavioral Homophily in Social Media via Inverse Reinforcement Learning: A Reddit Case Study. In . ACM, New York, NY, USA, 14 pages. https://doi.org/10.1145/nnnnnnn.nnnnnnn

## 1 Introduction

Social media platforms have become integral to modern society, shaping public discourse and influencing information flow. They have far-reaching offline effects, even impacting financial markets, as illustrated by the Reddit community *r/wallstreetbets*, which played a crucial role in the GameStop Short squeeze in early 2021 [34]. While concepts like network effects and influence maximization [21] offer valuable macroscopic insights, they often fail to capture the nuanced individual-level behaviors within niche online communities. This work studies online users at their most granular level – their online interactions.

Homophily – the tendency for individuals to engage with others who possess similar characteristics – is a key driver in shaping the dynamics of online social media platforms. Homophily drives the formation of online interest groups and communities and can even play a role in the spread of online misinformation [7]. Traditional measures of homophily rely on follower networks – explicit information about who follows whom. Other approaches rely on quantifying the shared hashtags, which works well for platforms such as X/Twitter and Facebook, where the platform structure is centered on individual relationships and explicit social ties. However, these measures are inadequate for platforms such as Reddit, which is organized around topic-based communities known as *subreddits* without explicit social ties or follower relationships. While content-based homophily measures (approaches that measure the similarity of the content produced or consumed by users) can be applied to Reddit, they offer little insights for a platform already organized along topical themes. This underscores the need for an alternative method to analyze homophily in such environments, focusing on the nature of interactions rather than observable affiliations or consumed content. To address this gap, we propose using Inverse Reinforcement Learning (IRL) – a framework to infer a policy that explains an observed behavior – to study behavioral homophily based on users' observed actions on the platform.

### 1.1 Unique Challenges

The unique challenges faced in our work are summarized as follows: **Limitations of traditional homophily measures.** Existing homophily measures focusing on follower networks or hashtags are inadequate for platforms where interactions are not follower-based. Moreover, user anonymity on platforms like Reddit complicates analysis, as demographic data such as gender or age is unavailable. **Applying IRL to hierarchical data.** While IRL has been applied to uncover the reward functions behind user decisions [27], applying it to hierarchical data, such as Reddit's conversation structures, remains challenging. Designing compact state representations that reflect the complexity of user interactions while addressing data sparsity is still an open research problem.
**Linking topical interest and posting behavior.** While the user topical interest in Reddit is quite well understood given the thematical subreddit community structure, the connections between users of unrelated communities remain largely unexamined. In particular, the relationship between users who display similar behaviors on completely different topics and subreddits is underexplored, as most measures of homophily do not account for user posting behavior.

### 1.2 Our Contributions

To address these challenges, we propose an Inverse Reinforcement Learning (IRL) framework for studying behavioral homophily, making the following key contributions:
**An IRL framework for analyzing user behavior.** We develop an IRL model tailored to social media platforms with hierarchical, forum-like data structures. The model defines state and action spaces that capture key features, such as the agreement in replies, encoding user activity and community-triggered interactions.

*Conference'17, July 2017, Washington, DC, USA*
2024. ACM ISBN 978-x-xxxx-xxxx-x/YY/MM
https://doi.org/10.1145/nnnnnnn.nnnnnnn

**A new measure of behavioral homophily via IRL.** We introduce a novel measure of behavioral homophily derived from the inferred policy map of our IRL framework. We contrast this measure with the commonly used topic homophily and validate its robustness using statistical significance tests, identifying significant behavioral differences across various online communities.

**Reddit case study.** We conduct a detailed case study on Reddit, analyzing subreddits focused on news, political ideology, human rights, and sexual identity. This analysis provides insights into connections between topical and behavioral homophily on Reddit, shedding light on what drives users' interactions.

### 1.3 Related Work

Understanding human behavior to uncover the underlying reward mechanisms of decision processes gained significant attention after the seminal work of Ng and Russell [32]. When combined with entropy regularization and deep learning, Inverse Reinforcement Learning (IRL) has evolved into a powerful tool for analyzing complex behavioral patterns, such as overtaking maneuvers in driving [45] or identifying optimal NHL players for fantasy sports [28]. While IRL has been extensively applied in vision-based domains, its application to social media has been more limited due to challenges in encoding the underlying data structure and ensuring sufficient data availability.

Early studies applying IRL to social media explored how feedback influences personal engagement on Reddit, showing that users tend to continue engaging based on the reception of their contributions [10]. Luceri et al. [27] applied predictive modeling to detect troll behavior on X (formerly Twitter) by identifying key behavioral features. Geissler et al. [17] examined propaganda strategies following the Russian invasion of Ukraine using a comparable framework. On YouTube, Hoiles et al. [20] leveraged IRL to model and predict viewer commenting behavior, demonstrating how the rational inattention model [41] can explain variations in user engagement. Among these platforms–X, YouTube, and Reddit–Reddit stands out for its highly hierarchical data structure, organized around nested discussions. Our framework uniquely adapts deep IRL by designing states, actions, and features that reflect Reddit's hierarchical conversation structures, considering platform-specific behaviors such as creating threads or root comments.

As our study focuses on homophily in social media behavior, it connects closely to research examining social media dynamics. Massachs et al. [29] investigated the roots of Trumpism on the subreddit *r/The_Donald* through the lens of homophily, social influence, and social feedback. In their study, homophily was measured through vector participation across different subreddits, which, while suitable for that case, lacks broader generalizability and behavioral detail. Monti et al. [31] evaluated homophily and heterophily among ideological and demographic groups in Reddit's *r/news* community, finding that users tend to engage with opposite ideological sides, while demographic groups, particularly age and income, exhibit homophily. This challenges the echo chamber narrative and highlights the role of affective polarization in a divided society. Other studies [12, 15] have challenged the echo chamber narrative on Reddit, showing that political interactions involve significant cross-cutting engagement, with polarization and hostility more prevalent within

political groups or asymmetrically between supporters, rather than between opposing sides.

Our work builds upon these insights by extending the analysis to a diverse set of subreddits, each with unique conversational patterns. We offer a deeper understanding of homophily and user behavior across various communities on Reddit by introducing a novel behavioral homophily measure through our IRL framework.

## 2 Preliminaries

In this section, we provide the necessary background on homophily, inverse reinforcement learning, and the structure of Reddit.

### 2.1 Homophily

Homophily—the tendency for individuals to associate with others who are similar [30]—plays a crucial role in shaping social networks. It is typically classified into *status homophily* and *value homophily* [26]. Status homophily occurs when ties form based on demographic or socioeconomic characteristics like age, race, or education, while value homophily is driven by shared beliefs, attitudes, and behaviors. These patterns influence not only who connects but also the overall structure and dynamics of the network [14]. In online networks, these tendencies can create clusters or echo chambers where users interact primarily with like-minded individuals, amplifying polarization and group identity [29].

Traditional measures, such as shared interests or topical similarity, have been widely used in various applications, mostly relating to political ideology. Colleoni et al. [9] investigates political homophily on Twitter/X using content based classifiers and social network analysis to infer affiliation to American political parties. Ram et al. [36] investigates the inference of users' political ideology through three homophilic lenses in lexical similarity, shared hashtags, and reshared content on Twitter/X. These measures help explain the formation of social ties but often miss the complexity of user behavior. For example, Aiello et al. [1] found that topical similarity predicts social links with up to 92% accuracy. However, Bisgin et al. [5] demonstrated that interest-based homophily alone does not fully explain new tie formation across platforms like Blog-Catalog, Last.fm, and LiveJournal, signaling the need to consider more nuanced behavioral factors.

Behavioral homophily–users feeling closer to those who behave similarly online–offers deeper insights into social dynamics. Figeac and Favre [16] showed that frequent interactions such as liking and commenting strengthens ties, especially among weak connections. Similarly, Pan et al. [33] demonstrated that social network homophily, using graph convolutional networks, improves user attribute predictions, even with limited data.

Understanding behavioral homophily is crucial for analyzing network cohesion and societal impact. It offers insights into user interactions and content exposure, with implications for improving recommendations, mitigating polarization, fostering inclusive networks, and designing effective interventions [23, 39].

### 2.2 Inverse Reinforcement Learning

Inverse Reinforcement Learning (IRL) offers a framework to infer the underlying motivations or reward structures that drive observed actions [32, 38]. In contrast to Reinforcement Learning (RL), which

learns how to optimize a known reward function, IRL focuses on inferring a reward function that explains observed behavior. The central problem of IRL is to deduce the latent preferences of an agent from observed state-action trajectories. In the context of online behavior, IRL aims to uncover the rewards users are implicitly maximizing based on their publicly observable actions. This methodology has been applied to differentiate normal user behavior from that of trolls–users who intentionally provoke or disrupt discussions [27].

IRL operates within the framework of a Markov Decision Process (MDP), where the reward function is unknown. The MDP is defined by the tuple $(\mathcal{S}, \mathcal{A}, P, \gamma, \tau)$, where $\mathcal{S}$ is a finite set of states, $\mathcal{A}$ is a finite set of actions, $P$ is the transition kernel, and $\gamma$ is the discount factor. Trajectories or demonstrations $\tau$ represent the observed state-action pairs over time. We employ the maximum entropy IRL framework [46] and its deep IRL extension [44], where the reward function is parameterized by a neural network. The reward function for state $s$ is learned as $R(s) = \boldsymbol{w}^\top \varphi_l(s)$, where $\varphi_1(s) = \sigma(W_1 s)$, and $\varphi_j(s) = \sigma(W_j \varphi_{j-1}(s))$ for $j \in \{2, \dots l\}$. Here, $\boldsymbol{w}$ is a weight vector, $\sigma$ the activation function, and the neural network is defined by the weights $W_j$ for $l$ layers, with all parameters collectively represented as $\boldsymbol{\theta} = \{W_1, \dots, W_l\}$. The reward function is optimized by maximizing the likelihood of the observed trajectories under a maximum-entropy framework. The policy is updated using soft Q-learning, and the neural network parameters are adjusted through backpropagation until convergence (see Algorithm 1).

---

**Algorithm 1** Maximum-Entropy Deep IRL

---

**Require:** State space $\mathcal{S}$, action space $\mathcal{A}$, discount factor $\gamma$, convergence threshold $\epsilon$, observed demonstrations $\tau$
**Ensure:** Optimal policy $\pi^*$, optimal reward function $R^*$
1: **while** not converged **do**
2:     **Update Reward Function**
3:     $R \leftarrow \text{NN}(\boldsymbol{\theta})$
4:     **Update Policy using Soft Q-Learning**
5:     $\pi \leftarrow \text{Soft Q-Learning}(\mathcal{S}, \mathcal{A}, \gamma, R, \epsilon)$
6:     **Compute Maximum Entropy Gradients**
7:     Compute $\frac{\partial \mathcal{L}}{\partial R}$ using state-action distribution from $\tau$
8:     **Update Neural Network Weights**
9:     Backpropagate gradients and update NN weights $\boldsymbol{\theta}$
10: **end while**
11: **Return** $R^*$, $\pi^*$

---

For further details, we refer the reader to Arora and Doshi [2], which provides an extensive survey of IRL methods.

## 2.3 Reddit

Reddit is a social media platform and is the 6th most visited website in the world (as of August 2024) [40]. The platform revolves around user-created communities focused on specific topics or interests called *subreddits*. Users personalize their content feed by following subreddits, unlike other platforms where users follow individuals.

Subreddits are user-moderated, with each defining its own rules and conduct guidelines, resulting in varied community dynamics across the platform. The content structure of Reddit is hierarchical

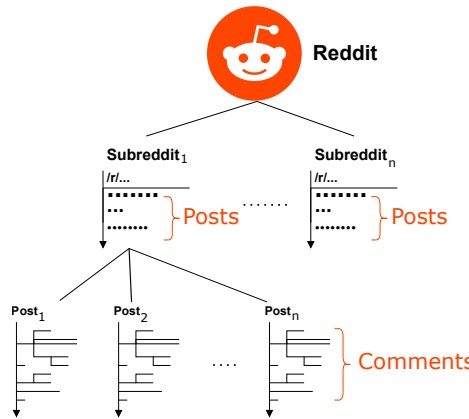

**Figure 1: Hierarchical structure of Reddit. Reddit is divided into numerous subreddits, with each subreddit consisting of posts. Each post contains its own comment section with each comment having its own comment tree.**

(see Fig. 1); the platform is first divided into subreddits, under which users may create and browse *posts* (also referred to as *threads* or *submissions*) that initiate threaded discussions. Posts can contain text along with media such as hyperlinks, images, or videos. Within each post, users engage in discussions through *comments*, which are text-based and can be nested to form conversation threads.

Reddit uses a voting system called "karma" to rank posts and comments. Upvotes and downvotes determine the visibility of content, with highly upvoted items gaining prominence, while those with negative karma are hidden. While a user's overall karma is displayed on their profile, it does not affect functionality.

The default content feed shows highly upvoted posts from subscribed subreddits. Additional views include *r/popular*, which highlights popular posts across Reddit, and *r/all*, which displays all posts, including potentially inappropriate (labeled NSFW) content.

## 3 Methodology

In this section, we introduce the framework for constructing a measure of behavioral homophily. Additionally, we outline the steps for developing a topic-based homophily measure, a standard approach in social network analysis, which we use in tandem with the proposed method in our case study. Fig. 2 provides a visual summary of the process, which is further elaborated in the subsequent sections. The process follows these steps: (1) subreddit selection, (2) user selection, (3) data collection, (4) data labeling, (5) policy learning via IRL, and (6) homophily inference.

Before sampling, our raw data consists of 1.3 TB of compressed text covering the entirety of Reddit during the period from January 1, 2015, to January 1, 2022. This data was collected using the pushshift Reddit API [3].

## 3.1 Subreddit Selection

Our objective is to develop a general measure of homophily that can be applied to diverse user groups with varying activity levels and engagement in controversial discussions, which may influence the degree of homophily or heterophily (anti-homophily) within these groups. To ensure the robustness and generalizability of our

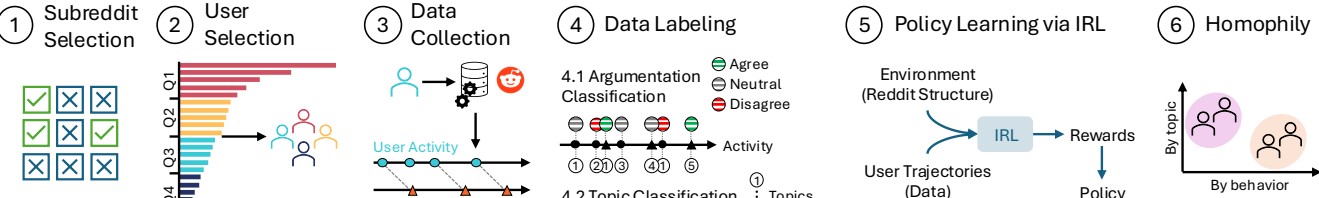

**Figure 2: Behavioral homophily inference framework for hierarchical social network data via Inverse Reinforcement Learning.**

approach, we strive to select users with diverse views who interact with a multitude of subreddits. However, while Reddit is organized along subreddits, a user's subreddit subscriptions are not publicly available on Reddit. This makes it impossible to determine which users are associated members of which subreddits, given any user can post in any subreddit. The mere presence or activity in a subreddit does not indicate how invested they are in that subreddit, as there may be other subreddits on which they are more active. Setting an arbitrary threshold for associating a user with a subreddit is not intuitive due to the difference in total activity levels between users. To address this issue, we introduce the concept of a user's *home subreddit* defined as the subreddit where the user is most active based on their comments across all of Reddit.

To find the subreddits with dedicated and active user bases, we first select a seed set of subreddits to determine those with the most home users. This seed set covers a wide range of topics ranging from general interest subreddits (e.g., *r/news*) to more controversial or niche communities (e.g., *r/The_Donald*). More detailed information on each subreddit is provided in Appendix A.

## 3.2 User Selection

We implemented a multi-step selection process to capture a representative sample of users across the entire timeframe. First, we compiled an initial user set based on activity levels across each subreddit in the seed set. For each year, we rank users in each subreddit by their activity, and we include the top 50 most active users in the initial set. To capture a range of activity levels, we divided the yearly rankings into quartiles and randomly sampled 50 users from each quartile. This results in 250 users per subreddit per year. Next, we examined an intermediary subset of 6,000 users, selected by randomly sampling one-third of the initial set. This step was necessary because determining a user's home subreddit requires analyzing their activity across the entire platform, a time-intensive process. From this subset, which contained 1,331 unique home subreddits, we focused on the 15 subreddits with the most home users for our case study, shown in Table 1. We then sampled 45 users from each of the 15 home subreddits, matching the size of the smallest subreddit in the group. After filtering out banned users and those with deleted accounts, the final dataset comprised 662 users.

## 3.3 Data Collection

We extract all user activity from Reddit for each individual in our sample. Direct *user activity* refers to actions explicitly initiated by users, such as creating threads, root comments, and replies. In addition, we capture indirect activity (*first-order response*) in

**Table 1: Subreddits examined in our case study.**

| Subreddit | Description |
|---|---|
| *r/AskReddit* | A platform for users to pose open-ended questions to the Reddit community. |
| *r/AsianMasculinity* | Supportive space for Asian men to discuss societal and dating challenges. |
| *r/aznidentity* | Activist community promoting Pan-Asian identity and opposing anti-Asian racism. |
| *r/Conservative* | Forum for discussing conservative politics and news. |
| *r/leagueoflegends* | Discussions about gameplay, strategies, and news for the video game League of Legends. |
| *r/memes* | Sharing internet memes and humorous content. |
| *r/MensRights* | Exploring issues related to men's rights and societal roles. |
| *r/Minecraft* | Community for Minecraft players and enthusiasts. |
| *r/news* | News articles about current events worldwide for discussion. |
| *r/NoFap* | Peer support forum for porn addiction and compulsive sexual behavior with a focus on abstinence. |
| *r/politics* | Discussion of current political events and opinions. |
| *r/soccer* | All topics related to association football: news, results, and discussions. |
| *r/teenagers* | Discussions relating to being a teenager. |
| *r/The_Donald* | Former subreddit supporting Donald Trump; banned for policy violations. |
| *r/worldnews* | Major global news excluding US internal news. |

the form of interactions triggered by the user's actions within Reddit's hierarchical structure (cf. Fig. 1). This structure, modeled as a directed acyclic graph, consists of parent-child relationships where each action can have multiple descendants. For this analysis, indirect activity is restricted to the first descendant (or "child") directly connected to the user's action.

Both direct and indirect activities are integral to modeling the conversational dynamics in which inverse reinforcement learning (IRL) is applied. In cases where users have deleted their accounts or their posts have been suspended, such content is marked as unavailable. This missing data, which may introduce noise or outliers in constructing the topic homophily baseline, is systematically handled during preprocessing by omitting the affected users.

## 3.4 Data Labeling

As highlighted in prior research, debates within the online landscape are dynamic and continuously evolving, with consensus formation closely tied to the arguments shared by individual users. Emotions, particularly emotionally charged content, contribute to polarization [8], pushing users toward more extreme positions. While linguistic features such as word choice and syntax are useful for detecting polarization, they are not its primary drivers. Instead, network attributes like echo chambers reinforce existing beliefs and limit exposure to opposing viewpoints, further shaping the trajectory of discussions [13].

However, consensus-building relies fundamentally on the elements of agreement, disagreement, and neutrality, which are the core drivers of how discussions unfold. These classifications are commonly employed in opinion mining and argumentation theory to analyze online discourse, also known as argumentation (stance) classification [25]. We categorize each comment into three categories–"agree," "neutral," or "disagree"–to better understand user interactions (see below for technical details). Incorporating this classification as an additional feature in our dataset enables deeper analysis and provides insights into why users engage in discussions on social media platforms. Furthermore, individual motivations and decision-making processes may vary across communities, as these dynamics are often subreddit-specific.

**Argumentation Classification.** For our classifier, we fine-tune a pre-trained DeBERTaV3 model [19] using the *DEBAGREEMENT* dataset [35]. This dataset consists of labeled comment-reply pairs from five subreddits: *r/BlackLivesMatter*, *r/Brexit*, *r/climate*, *r/democrats*, and *r/Republican*, with each pair labeled as "agree," "neutral," or "disagree." We selected the DeBERTaV3 model due to its strong performance across a range of natural language processing tasks, and because it shares the same model lineage as BERT, which was used in [35]. For each input, the parent and reply text were concatenated, and the model was trained to classify the interaction into one of the three categories.

**Topic Classification for Topic-based Homophily.** We implement a baseline, topic-based homophily based on users' discussion topics. We use a pre-trained BERTopic model. BERTopic [18] employs a transformer architecture combined with a class-based term frequency-inverse document frequency (c-TF-IDF) weighting scheme to generate a set of $K$ topics. From our sample of posts, we derive $K = 484$ distinct topics. Using these topics, we construct a topic-based homophily measure, where each user's activity is represented by a vector describing how frequently they communicated within each topic over the observation period.

We provide further implementation details for each classification step in Appendix A.3.

## 3.5 Policy Learning via IRL

To represent user interactions within Reddit, we must define an IRL framework within which we operate. We define the user as the agent operating within an environment that encapsulates the entirety of the Reddit platform, excluding the user themselves. Therefore, each user agent is independent and does not directly interact with other agents (they can interact indirectly, mediated by the environment). We use maximum entropy deep inverse reinforcement learning to recover a reward function based on a trajectory of constructed state-action feature pairs. The user's trajectory is constructed from the stream of events that involve the user across all of Reddit, which we map into state-action feature pairs. We define the following state features:

- *Initial thread (IT).* First or only interaction, creating a new thread.
- *Initial root comment (IRC).* First or only interaction, posting a root comment.
- *Initial reply (IR).* First or only interaction, replying to a comment, further split into agreement ($IR_+$), neutrality ($IR_\smile$), and disagreement ($IR_-$).

- *Engaged root comment (ERC).* Already interacted, posting a root comment.
- *Engaged reply (ER).* Already interacted, replying to a comment, further split into agreement ($ER_+$), neutrality ($ER_\smile$), disagreement ($ER_-$).
- *Get reply (GR).* Receiving a reply on any reply or comment, further split into agreement ($GR_+$), neutrality ($GR_\smile$), disagreement ($GR_-$).

In summary, this results in 12 states, with the agent always starting in one of the three initial states. At each timestep, having observed the state, the agent takes one of the following 6 actions, which influences the next state the agent transitions to:

- *Wait reply (WR).* User waits for a reply to one of their comments.
- *Create new thread (CT).* Start a new discussion in the subreddit.
- *Post root comment (RC).* Directly comment on the thread's original post.
- *Post reply comment (PR).* Respond to another user's comment, creating a nested conversation. We further dissect this state between agreement ($PR_+$), neutrality ($PR_\smile$), disagreement ($PR_-$).

We infer the user's policy $\pi_u$ from the user's reward function using value iteration. This policy can be represented as a $12 \times 6$ matrix, where each row corresponds to the action distribution given a state.

## 3.6 Homophily Inference

After constructing a policy $\pi_u$ for each user $u$ via Inverse Reinforcement Learning (IRL) (see Sections 2.2 and 3.5), we quantify user homophily by analyzing the behavioral similarity of users.

**Behavioral Homophily.** We state that two users have high behavioral homophily when their inferred policies are similar. We introduce the *Symmetric Weighted Kullback-Leibler Divergence* (SWKL). This measure extends the standard Kullback-Leibler (KL) divergence [24] by incorporating visitation weights, assigning higher importance to states that are frequently visited by each user individually and down-weighting states that are rarely visited. This weighting reduces the impact of noise from infrequent states, ensuring that divergence is dominated by states where the user's behavior is more representative.

Each user's behavior is characterized by a policy describing their action distributions over states. Let $\mathcal{U} = \{1, \ldots, U\}$ denote the set of users, where $U$ is the total number of users and $u \in \mathcal{U}$. Formally, let $\pi_u$ represent the policy of user $u$ over a finite set of states $\mathcal{S}$ and actions $\mathcal{A}$. For a given state $s \in \mathcal{S}$, $\pi_u^s$ is the distribution of actions taken at state $s$. The policy is inferred using IRL from the user's trajectory, $\tau_u = \{(s_1, a_1), \ldots, (s_{|\tau_u|}, a_{|\tau_u|})\}$, which records the sequence of states visited and actions taken by the user. To account for how often each state is visited by user $u$, we define the state weight as $w_u^s = \left( \sum_{k=1}^{|\tau_u|} \mathbb{1}_{\{s_k=s\}} \right) / |\tau_u|$, where $\mathbb{1}_{\{s_k=s\}}$ is the indicator function, taking the value 1 if the state $s_k = s$ and 0 otherwise. This weight reflects the proportion of time user $u$ spends in state $s$. The Symmetric Weighted Kullback-Leibler Divergence (SWKL) between two users $u$ and $u' \in \mathcal{U}$ is then defined as

$$\text{SWKL}(\pi_u, \pi_{u'}) = \frac{1}{2} \sum_{s \in \mathcal{S}} \left( w_u^s D_{KL}(\pi_u^s \| \pi_{u'}^s) + w_{u'}^s D_{KL}(\pi_{u'}^s \| \pi_u^s) \right),$$

where $D_{KL}(\cdot \| \cdot)$ denotes the KL divergence between two probability distributions. By symmetrizing and weighting the divergence,

SWKL provides a more balanced and robust measure of behavioral similarity, emphasizing the most representative states for each user while reducing sensitivity to rare state visits.

**Topic Homophily.** We further construct a baseline topic homophily using the topic vector $\mathbf{v}_u$ obtained from the topic classification (Section 3.4). Two users have a high topical homophily if they emit messages about similar topics. To quantify the similarity of topics among users, we use cosine distance as our measure of topic homophily. For each user $u$, we construct a topic vector $\mathbf{v}_u$ by using the number of posts assigned to each of the 484 topics identified by BERTopic. The cosine distance between two users $u$ and $u'$ is given by

$$\cos(\mathbf{v}_u, \mathbf{v}_{u'}) = 1 - \frac{\mathbf{v}_u^\top \mathbf{v}_{u'}}{\|\mathbf{v}_u\| \|\mathbf{v}_{u'}\|}.$$

This provides a measure of topic alignment between users based on the distribution of their posts across topics.

## 4 Case Study

This section presents our Reddit case study, focusing on subreddit-specific criteria that capture the distinctive dynamics of group conversations and individual user contributions. Instead of relying solely on general homophily principles, we investigate the nuanced interactions unique to each subreddit. We demonstrate that examining topics alone in a thematically organized platform like Reddit provides limited insights. By incorporating user behavior–through analysis of user policies–we reveal that homophily manifests differently across topical and behavioral dimensions. Our analysis centers on home subreddits (as introduced in Section 3.2), representing the subreddits with the most active primary commenters.

### 4.1 Homophily Across Subreddits

We explore homophily across users' home subreddits along two dimensions: topic and behavior (policy). Specifically, we examine whether users who share the same primary subreddit exhibit similar topical and behavioral patterns within and across their broader activity on Reddit.

**Subreddit Topic Homophily.** We assess topic homophily by asking: "If two users primarily engage with the same subreddit, how closely do their activities across Reddit align in terms of topics?" To measure this, we compute the mean cosine distance between user pairs across different subreddits. Formally, let $C \subset \mathcal{U}$ and $C' \subset \mathcal{U}$ represent the home users two distinct subreddits, where $C \cap C' = \emptyset$. The mean cosine distance between these two subreddits is

$$\overline{\cos}(C, C') = \frac{1}{|C||C'|} \sum_{u \in C} \sum_{u' \in C'} \cos(\mathbf{v}_u, \mathbf{v}_{u'}).$$

Fig. 3a presents the mean cosine distance across 15 subreddits. Most subreddits exhibit the strongest topical homophily within themselves, indicated by the diagonal, which shows that users who post primarily in the same subreddit are more likely to engage in similar topics across Reddit.

Notably, strong overlap is observed between *r/AsianMasculinity* and *r/aznidentity*, likely due to their shared focus on Asian identity discussions in the US and the Western world. Another cluster with substantial topic overlap includes *r/news*, *r/worldnews*, *r/politics*, *r/Conservative*, and *r/The_Donald*. We attribute this to their shared

focus on broadly defined political events, particularly US politics. Within this group, *r/worldnews* shows weaker overlap, likely due to its policy of excluding US internal news, resulting in less topical alignment. The overlap observed in the other subreddits–*r/news*, *r/politics*, *r/Conservative*, and *r/The_Donald*–stems from their predominant focus on American politics. Given that subreddits are thematically organized, it is unsurprising that users' topics align with the primary subreddit they engage in.

**Subreddit Behavioral Homophily.** We examine how behavioral homophily, as reflected in user policies, aligns with home subreddits. Using the same method as for topic homophily, we compute the mean SWKL between users from different subreddits. Formally, the mean SWKL between two subreddits $C$ and $C'$, representing their home users, is expressed as

$$\overline{\text{SWKL}}(C, C') = \frac{1}{|C||C'|} \sum_{u \in C} \sum_{u' \in C'} \text{SWKL}(\pi_u, \pi_u').$$

Fig. 3b illustrates the mean SWKL across 15 subreddits. Compared to topic homophily, user policy shows weaker alignment with home subreddits, with most subreddits displaying overlap with others and none being entirely unique in behavior. This is intuitive, as users can deploy similar behaviors around very different topics (and subreddits are topically defined).

Two distinct sets of subreddits exhibit substantial internal policy overlap. The first set covers topics such as politics and activism and includes *r/worldnews*, *r/news*, *r/politics*, *r/soccer*, *r/leagueoflegends*, *r/Conservative*, and *r/MensRights*. The second group covers gaming and youth topics, and consists of *r/NoFap*, *r/memes*, *r/teenagers*, and *r/Minecraft*. The lack of overlap between these groups makes sense, as they serve very different purposes and cater to different cohorts, which in turn exhibit different behaviors. Interestingly, *r/NoFap* and *r/memes* shows weaker internal policy alignment, suggesting greater user behavioral diversity. Additionally, we find that topic and policy homophily can diverge. For example, *r/AsianMasculinity* and *r/aznidentity* demonstrate strong topical overlap but weak policy similarity, indicating that while users discuss similar subjects, their user posting behaviors vary substantially.

### 4.2 Behavioral Personas Across Reddit

Here, we explore whether users can be grouped solely based on their behavior. We apply *k-means* clustering to user policies, selecting $k = 5$ based examining the tradeoff between the silhouette score and the gap statistic for various values of $k$ (see Appendix B for an in-depth analysis).

**Five Behavioral Personas.** We interpret each of the obtained clusters as a behavioral persona, and Fig. 3c summarizes the action composition across each cluster:

*Thread Creators* (25 users) focus on creating new threads rather than engaging with existing content; they have a high probability for the *CT* action (see Section 3.5). *Example:* user posts a question and does not interact with the thread any further.

*Root Only* users (114 users) primarily interact with root posts by posting first-level comments. They have minimal engagement with other replies. *Example:* user answers the questions in the post but does not engage in any other way with the thread.

*Root Favored* users (263 users) – similar to Root Only, users prefer to reply to root comments; however, they occasionally post

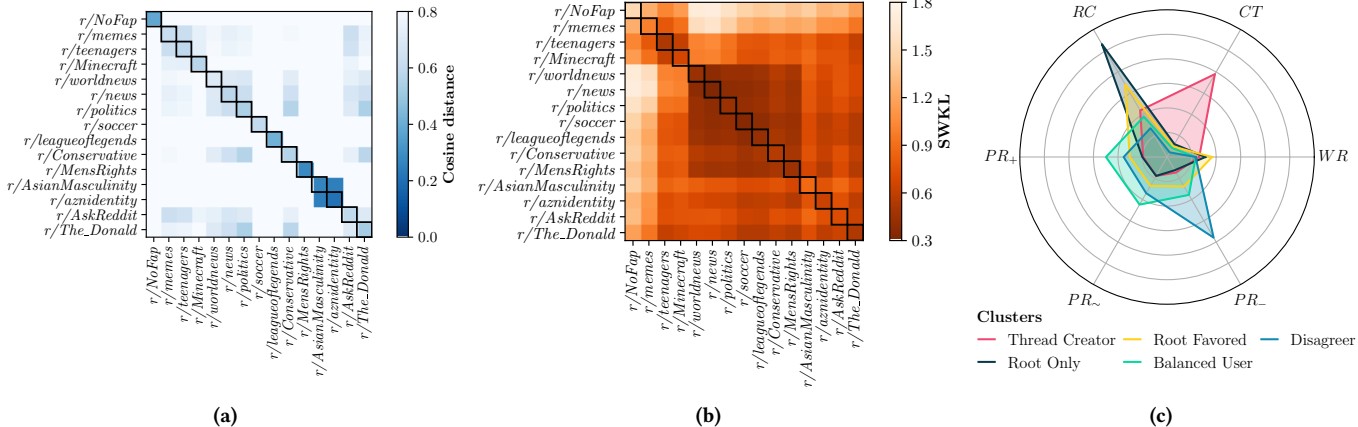

|  |  |  |
|---|---|---|
| (a) | (b) | (c) |

**Figure 3: The mean similarity between pairs of subreddits, with darker colors indicating greater similarity: (a) topical (cosine distance) similarity and (b) behavioral (SWKL) similarity. (c) Cluster action composition.**

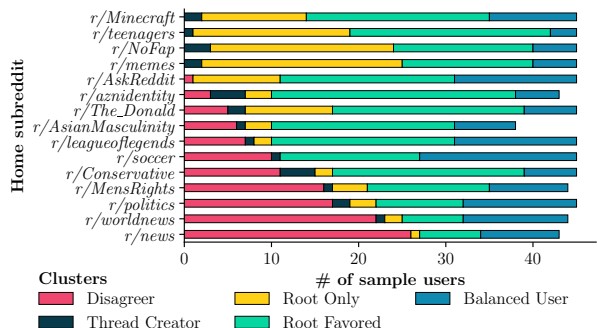

**Figure 4: Policy persona composition of each subreddit.**

replies without a preference for agreement ($PR_+$), neutrality ($PR_\sim$), disagreement ($PR_-$).

*Balanced Users* (136 users) display a balanced approach, with a slight preference for replies over root comments.

*Disagreers* (124 users) frequently post disagreeing replies, especially in response to disagreement; interestingly, they do not seek to engage in discussions beyond their disagreeing reply as they do not wait for additional replies (low $WR$ action).

**Posting and Reacting to Content vs. Disagreers.** Fig. 4 shows the distribution of these personas across subreddits, revealing significant variability. Overall, 57% of users are classified as either "Root Favored" or "Root Only," preferring root posts over discussions. Subreddits with a focus on political discussion, such as *r/news*, *r/worldnews*, and *r/politics*, have a higher proportion of "Disagreers." Interestingly, *r/The_Donald*, despite its political focus, has a low proportion of "Disagreers" (5 out of 45 users). A qualitative review of 100 comments reveals that while much of the content is abusive or hateful, users tend to agree, targeting hate toward specific individuals rather than engaging in debate against each other.

In contrast, subreddits like *r/memes*, *r/NoFap*, *r/teenagers*, and *r/Minecraft* have no "Disagreers", reflecting their non-political nature. In subreddits without "Disagreers", there is a higher proportion of "Root Only" users, despite their varied themes. A qualitative review of 20 threads per subreddit reveals a common pattern: threads typically start with a meme or image, seeking validation rather

than extended discussion. "Thread Creators" are sparse and tend to focus on specific subreddits, often their home subreddit or closely related ones. For example, a frequent commenter in *r/leagueoflegends* primarily creates threads in *r/Lolboosting*, a subreddit for account boosting in League of Legends.

## 4.3 Homophily Across Home Users

We investigate the relationship between topical and behavioral (policy) homophily to answer the question: "Do topically aligned users exhibit similar behaviors?"

We analyze pairs of subreddits using the Spearman correlation test – a non-parametric test capturing both linear and non-linear relationships. Specifically, we test the pairwise SWKL values between users of subreddits $C$ and $C'$, defined as the set $\{\text{SWKL}(u, u') \mid u \in C, u' \in C'\}$, as well as pairwise cosine distances between users, represented as $\{\cos(\mathbf{v}_u, \mathbf{v}_{u'}) \mid u \in C, u' \in C'\}$. A 5% significance level is applied, with Bonferroni correction for multiple comparisons.

**Behavioral and Topical Homophily Mostly Agree.** The results, displayed in Fig. 5a, show that most subreddits exhibit a positive correlation (red), meaning that users with similar topical preferences tend to exhibit similar behaviors. Non-significant results are shown in gray. An exception arises between *r/leagueoflegends* and *r/soccer*, which show a negative correlation (blue). This suggests that, in these subreddits, the more similar users are behaviorally, the less likely they are to share topical interests.

**The *r/soccer-r/leagueoflegends* Anomaly.** We further explore the relation between *r/soccer* and *r/leagueoflegends* by comparing the topics discussed in each subreddit. We construct the topic vectors from 1.5 million randomly sampled comments using BERTopic, classifying the comments into one of the 484 topics extracted in data labeling (see Section 3.4). The cosine distance between the topic vectors for *r/leagueoflegends* and *r/soccer* is 0.944, indicating minimal thematic overlap between the discussion of the two subreddits. The anomaly clears when we consider the behavioral homophily. Fig. 6 plots users' cosine distance from their home subreddit topic vector (x-axis) against their SWKL from soccer users (y-axis). We observe that users more aligned with their home subreddit topics tend to have lower SWKL, suggesting that deeper engagement with a subreddit leads to behavioral convergence. We hypothesize that this

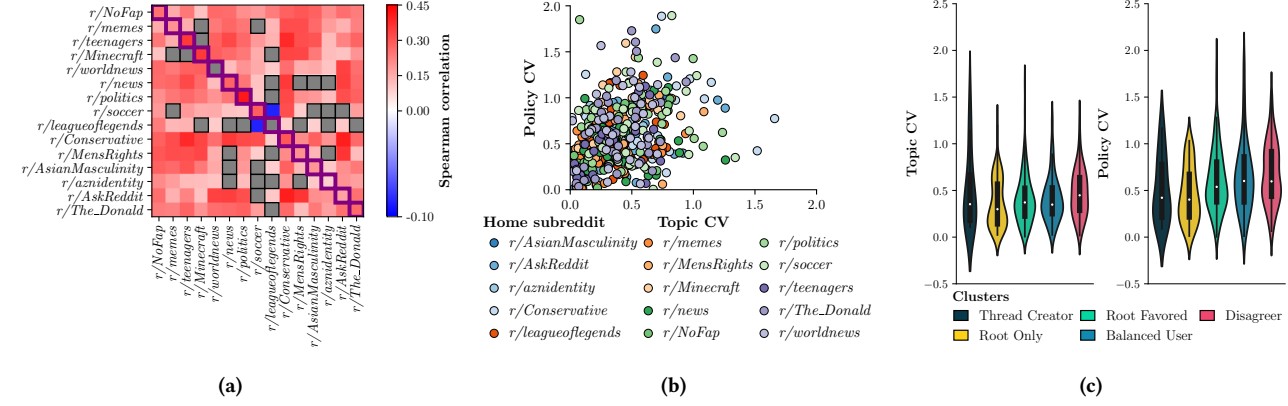

(a)

(b)

(c)

**Figure 5: (a) Spearman correlation between subreddit topic and behavioral homophily. Statistically non-significant results are indicated in gray. (b) Temporal stability of homophily by home subreddits. (c) Temporal stability of homophily by clusters.**

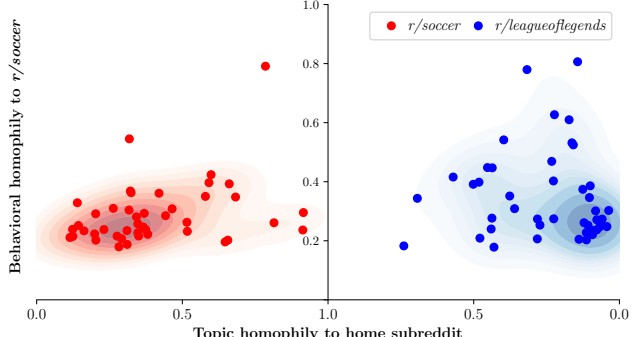

**Figure 6: Comparison of *r/soccer* and *r/leagueoflegends* users: Topical (Cosine) vs. behavioral (SWKL) distances. Smaller values indicate greater homophily and lower divergence.**

convergence arises from shared user characteristics between sports and e-sports communities, where users exhibit similar behaviors despite engaging with distinct topics.

### 4.4 Homophily Stability Over Time

To analyze the evolution of user behavior and topic interests over time, we partition each user's trajectory into annual segments. Let $\tau_u$ represent the complete trajectory of user $u$, and $\tau_{u,t}$ the trajectory for year $t$, where $t \in \{1, \ldots, T\}$ denotes the observation period. For each year, we calculate a user-based homophily measure, $h_u^m$, where $m \in \{\mathbf{v}, \pi\}$ corresponds to either topic-based homophily (using $\mathbf{v}$) or behavioral homophily (using $\pi$). We compute the change in homophily between consecutive years as $\Delta_{t,t+1}^{m,u} = h_u^m(t+1) - h_u^m(t)$, for $t \in \{1, \ldots, T-1\}$. To quantify the stability of homophily over time, we calculate the coefficient of variation (CV) as $CV_u^m = \frac{\sigma_u^m}{\mu_u^m}$, where $\sigma_u^m$ is the standard deviation and $\mu_u^m$ is the mean of the homophily changes $\{\Delta_{t,t+1}^{m,u}\}_{t=1}^{T-1}$. Since CV is a relative measure, variations in the scale of homophily are insignificant.

The comparison of temporal stability (CV) for both topic and behavioral homophily reveals no significant patterns (cf. Fig. 5b), with overall variability remaining consistent across subreddits. However, a closer examination of user roles (cf. Fig. 5c) uncovers more nuanced differences. "Thread Creators" exhibit lower Policy CV,

indicating stable behavior over time, but show greater variability in Topic CV, suggesting that while they regularly initiate discussions, their topic interests vary considerably. In contrast, "Disagreers" display higher variability in Policy CV, reflecting their more reactive and unpredictable engagement patterns.

These findings underscore the importance of incorporating both topic and behavioral homophily, as focusing solely on topics misses key aspects of user behavior. Behavioral dynamics, especially in relation to user roles and subreddit interactions, are essential in understanding user engagement patterns on Reddit.

### 5 Conclusion

In this study, we demonstrated that behavioral homophily can be inferred from hierarchical discussion data using inverse reinforcement learning. Our findings indicate that, across various user groups, the behavioral homophily measure closely aligns with traditional value-based (topic) homophily. Additionally, we highlighted the significant role that consensus mechanisms play in shaping user engagement within online discussions. This approach is particularly powerful for platforms with largely anonymous users, where traditional social network-based homophily measures—relying on explicit network features—are often unavailable. By facilitating more granular insights into individual user behaviors, this method offers a unique lens through which to analyze engagement patterns.

However, the approach does come with limitations. First, in the case of Reddit, the platform provided access to its complete uncensored hierarchical conversation structure, allowing us to examine both direct and indirect user activity. As many platforms increasingly restrict data access [11], applying this method universally becomes more challenging. Second, IRL is highly dependent on the size of the state and action space, requiring substantial amounts of data to avoid biased estimations. On platforms with sparse data, convergence to meaningful results is not guaranteed, limiting the approach's effectiveness in these contexts.

Finally, by analyzing a diverse selection of subreddits—including general, niche, and controversial content—we uncovered the intrinsic drivers motivating individuals to engage in online discourse. This analysis provided deeper insights into the dynamics of online communities and the underlying factors shaping user interaction and participation.

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

## A Dataset

In Section 3, we provided a brief overview of the steps used to construct our behavioral homophily measure from hierarchical Reddit data. In this section, we delve deeper into key aspects of the data selection process and offer insights into the data underlying this study. In Fig. 7, we visualize user activity on Reddit as a graph, highlighting the relationships between different subreddits.

### A.1 Extended Details on Subreddits

A comprehensive discussion of the topic selection is provided in Section 3.1, where a diverse set of subreddits is selected to sample from in Table 1. Below, we offer further insights into each subreddit in our initial seed set, focusing on their relevance to this study in terms of discussion dynamics and perceptions of controversial content.

*A.1.1 r/The_Donald.* *r/The_Donald* was a subreddit dedicated to Donald Trump and his supporters, created in June 2015 following Trump's announcement of his presidential campaign. It quickly became one of the platform's most active communities, playing a significant role in the alt-right movement surrounding Trump, particularly during his campaign and presidency. The subreddit was closely monitored by Trump's team due to its influence [37].

The community was known for creating and spreading media content, such as memes, that used humor and visuals to promote political messages. However, it also faced several controversies. Moderators and users actively manipulated Reddit's content algorithm to boost *r/The_Donald* posts on *r/all*, the platform's feed for all subreddit content, prompting Reddit to alter its algorithm in response [22].

*A.1.2 r/NoFap.* *r/NoFap* is a subreddit promoting abstinence from pornography and masturbation. The community has faced criticism for fostering sexist and misogynistic rhetoric, including the idolization of testosterone and masculinity, the objectification of women as rewards, and the shaming of sexually active women [6, 42].

*A.1.3 r/aznidentity and r/AsianMasculinity.* *r/aznidentity* and *r/AsianMasculinity* are subreddits centered on issues affecting Asian-American men and the broader Asian male diaspora in the Western world. Discussions frequently address the sexual emasculation of Asian men in Western culture, often accompanied by misogynistic undertones, including claims that Asian-American women in interracial relationships contribute to perpetuating this stereotype.

*A.1.4 r/Conservative.* *r/Conservative* is a subreddit centered on conservative ideologies, politics, and current events, primarily from a right-leaning perspective. The community presents itself as a platform for like-minded individuals to share news, opinion pieces, and engage in discussion. Conversations are largely focused on American politics, with far-right elements frequently present.

*A.1.5 r/MensRights.* The *r/MensRights* subreddit claims to focus on men's issues and advocate for gender equality. However, it has

been criticized for its anti-feminist and often misogynistic tone, as well as for promoting narratives that downplay or deny systemic gender inequalities faced by women.

*A.1.6 TwoXChromosomes.* *r/TwoXChromosomes* is a subreddit aimed at providing a supportive space for discussions focused on women's perspectives, experiences, and issues. However, the community has faced criticism for its moderation practices, where dissenting opinions are often downvoted, dismissed, or removed, potentially perpetuating a victimhood narrative and oversimplifying complex gender issues.

*A.1.7 r/communism.* The *r/communism* subreddit focuses on discussions, news, and perspectives related to Communist and Marxist political and economic ideologies. It has been criticized for its strict moderation policies and for promoting authoritarian regimes and ideologies.

*A.1.8 r/Antiwork.* The *r/Antiwork* subreddit is a community focused on discussions about working conditions and labor activism. Originally created to explore anti-work ideology within post-left anarchism, it has since expanded to encompass broader left-wing critiques of traditional work culture, with users advocating for reevaluating societal norms around work, labor, and capitalism. Moderators have expressed a vision for a society where people either don't need to work at all or have greatly reduced work obligations. The subreddit has faced criticism for promoting an overly simplistic view of work and productivity, and for endorsing and celebrating laziness.

*A.1.9 Non-Social-Political Subreddits.* To compare with the previously controversial communities, we also examine several non-social-political subreddits, including:

- *r/minecraft*: A subreddit focused on discussions about the open-world sandbox game Minecraft.
- *r/soccer*: A subreddit dedicated to the discussion of association football.
- *r/news* and *r/worldnews*: Two news-focused subreddits. *r/worldnews* differs in moderation by actively filtering US-centric news and US political content.
- *r/fuckcars*: A subreddit opposing car-centric lifestyles and the automobile industry, where users share memes, stories, and discussions on the negative societal and environmental impacts of car culture.

### A.2 Descriptive Statistics

After selecting the subreddits and users, we proceeded to collect data for the representative sample. This involved extracting all platform interactions relevant to constructing the state and action spaces for each user, including direct user activity and first-order responses. An overview of the state and action distributions is provided in Fig. 8 and Fig. 9.

**Remark.** Additionally, we recognize that certain platform interactions, such as upvotes or downvotes, are not included in our analysis, which may introduce a minor bias. While users can engage with content in these ways on Reddit, the available public data and the timing of conversation snapshots do not allow for

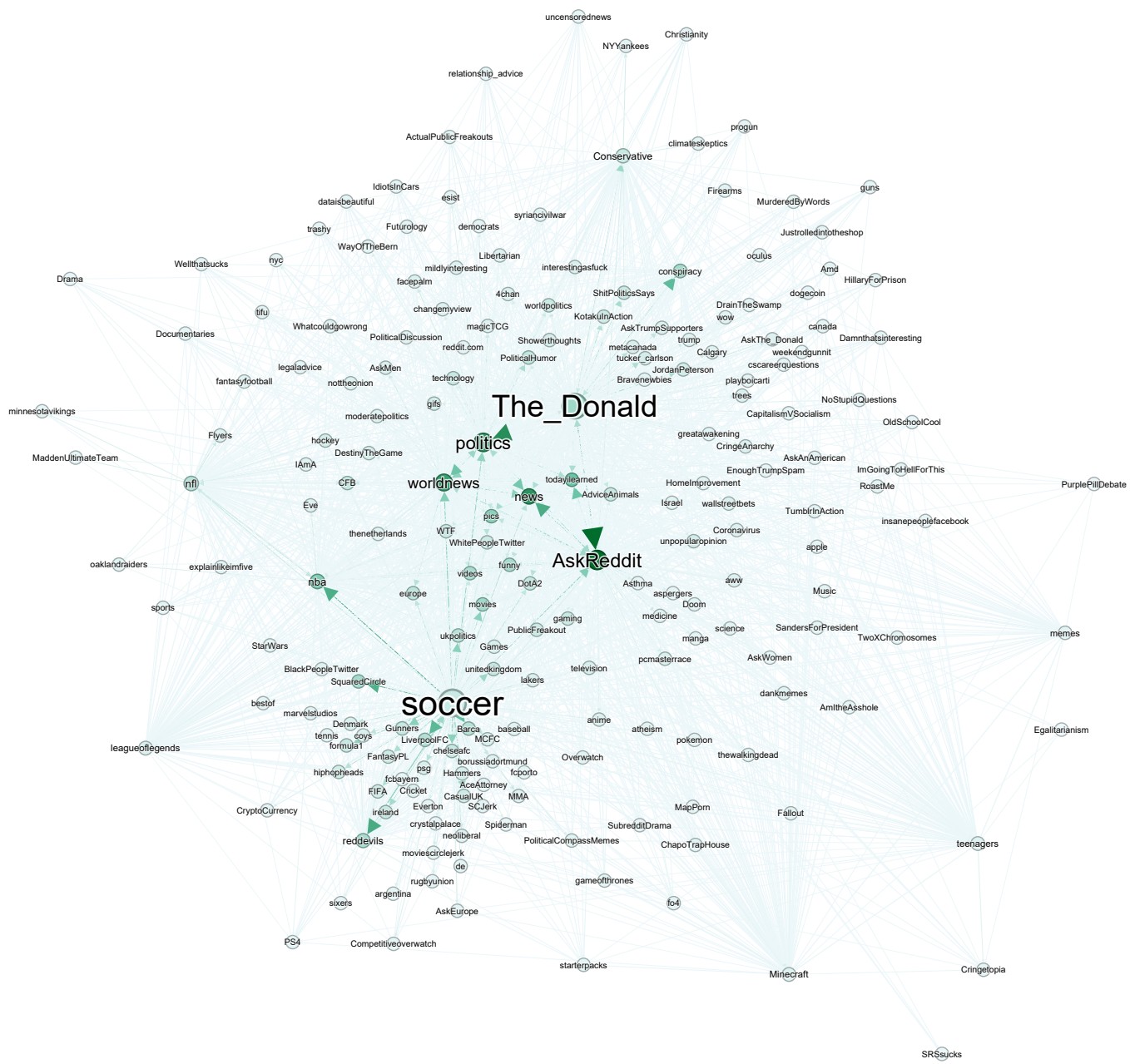

**Figure 7: Graph of user activity between subreddits on Reddit. Nodes represent home subreddits, with their size and labels indicating the amount of outward activity from home users. The color of each node reflects the activity it receives from other subreddits. Edges represent inter-subreddit activity, with the color and thickness of the edge head indicating the volume of activity.**

reliable tracking of when these actions occurred. As a result, we have excluded them from our analysis.

## A.3 Post Labeling

*A.3.1 Argumentation Classification.* We fine-tune a pre-trained DeBERTaV3 model for argumentation classification, as outlined in Section 3.4. Using the HuggingFace [43] implementation of De-BERTaV3, we fine-tuned the model with the pretrained weights "microsoft/deberta-v3-base". The model was optimized using the AdamW optimizer with a learning rate of $0.5 \times 10^{-4}$, a batch size of 8, and trained for 3 epochs with a warm-up of 500 steps. We applied an 8:1:1 training, validation, and test split to the *DEBAGREEMENT* dataset [35], comparing the performance of BERT, RoBERTa, and

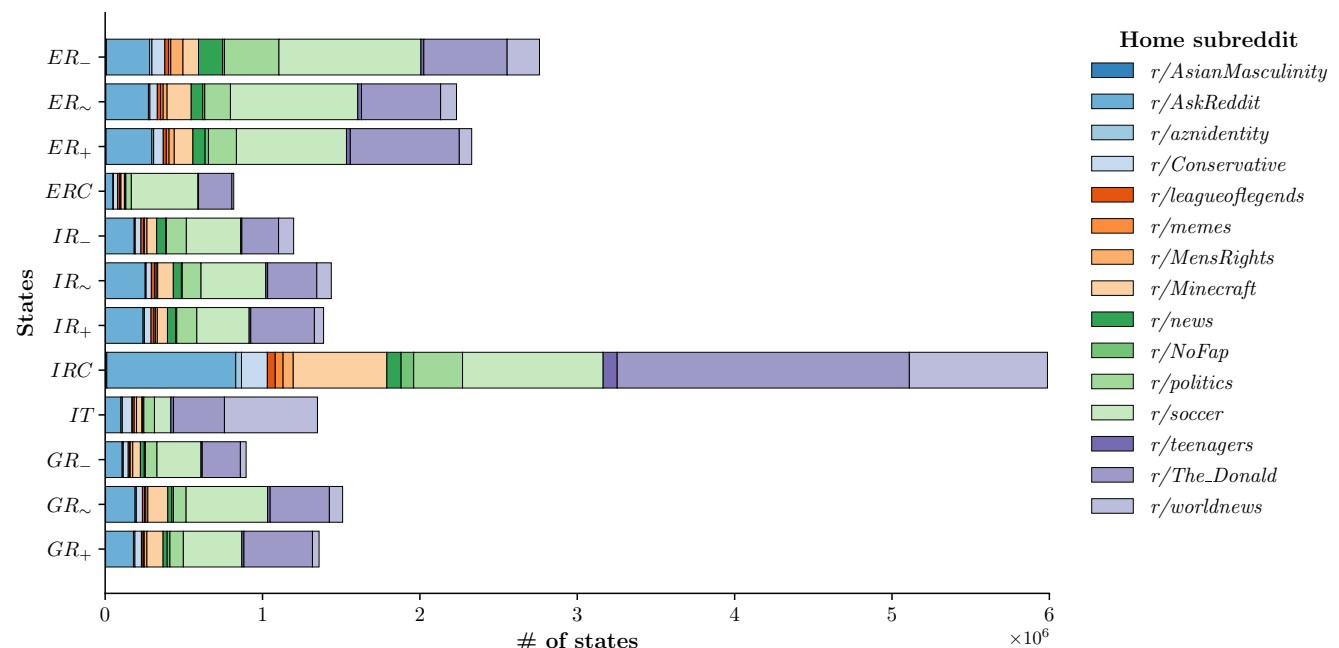

**Figure 8: User state visitation frequency, separated by home subreddit.**

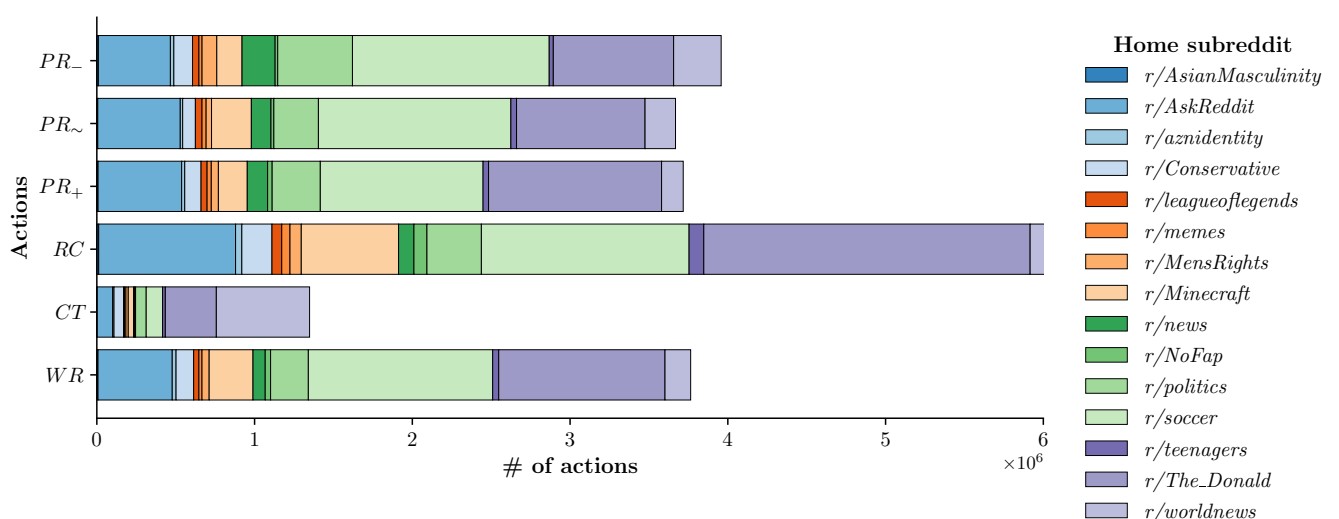

**Figure 9: User action visitation frequency, separated by home subreddit.**

DeBERTaV3. Unlike Pougué-Biyong et al. [35], we did not preserve the temporal order in our splits. Additionally, we explored other pre-trained models. The results, shown in Table 2, indicate that DeBERTaV3 outperforms both our own experiments with BERT and RoBERTa, as well as all reported results from Pougué-Biyong et al. [35].

A.3.2 *Topic Classification.* As described in Section 3.4, we use a pre-trained BERTopic model [18] to extract a set of $K$ topics for

| Pretrained Model | Accuracy | F1 |
|---|---|---|
| BERT | 0.666 | 0.664 |
| Roberta | 0.671 | 0.669 |
| DeBERTaV3 | **0.683** | **0.680** |

**Table 2: Empirical results for fine-tuning disagreement classification on the *DEBAGREEMENT* dataset. The highest scores are highlighted in bold.**

building our topic homophily baseline. We use the HuggingFace implementation of BERTopic for this task.

To create the document set for topic extraction, we collect all comments and submissions from our 662 users. Submissions are considered based solely on their title text, ignoring body content, images, videos, and links. We apply preprocessing by removing stop-words using NLTK [4] and filtering out empty, deleted, or removed titles and comments.

This results in a dataset of 5,910,728 documents, on which we apply BERTopic to extract topics. We set a minimum threshold of 1,000 documents per topic to limit the number of topics, yielding $K = 484$ distinct topics.

## B Behavioral Personas Clustering

To select the optimal value of $k$ for $k$-means clustering of user policies in Section 4.2, we used a combination of the Gap Statistic and Silhouette Score. The Gap Statistic compares the total within-cluster variation for different $k$ values to the expected variation under a uniform data distribution, while the Silhouette Score measures cluster separation, with higher scores indicating better-defined clusters. We explored $k$ values between 2 and 10, as shown in Fig. 10.

The two measures provided conflicting results: the Silhouette Score favored $k = 2$, while the Gap Statistic suggested $k = 10$. To reconcile this, we examined the largest drops in the Silhouette Score, aiming to select $k$ before the largest drop to preserve cluster separation. While the largest drop occurred between $k = 2$ and $k = 3$, such a small value of $k$ was not informative for our analysis. We instead considered the next largest drop, between $k = 5$ and $k = 6$.

To balance this with the Gap Statistic, we examined the delta of the Gap Statistic across values of $k$, using a threshold of 0.05 for the change. This threshold was met between $k = 5$ and $k = 6$ (see Fig. 11). Based on these findings, we selected $k = 5$ as a compromise between the two measures.

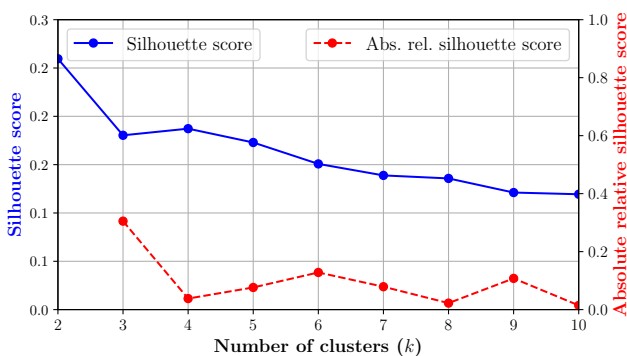

Figure 10: Silhouette score for $k$-means clustering.

## C Validation

Our method for measuring behavioral homophily employs maximum entropy deep inverse reinforcement learning (Deep-IRL). In Appendix C.1, we outline the hyperparameters used, followed by an analysis of the descriptive power of the inferred user policies

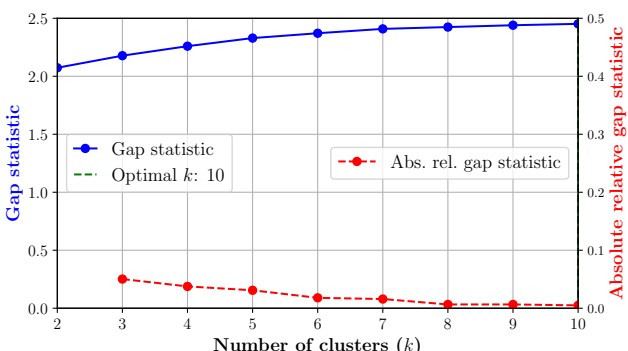

Figure 11: Gap statistic for $k$-means clustering.

(Appendix C.2) and insights into the symmetric weighted Kullback-Leibler divergence scores across subreddits (Appendix C.3).

### C.1 Hyperparameters for Deep-IRL

In this section, we summarize the key hyperparameters used in all experiments (see Table 3). For readers unfamiliar with Inverse Reinforcement Learning (IRL), we recommend consulting foundational IRL literature, as these hyperparameters differ in important ways from those typically used in more complex reinforcement learning tasks.

| Hyperparameter | Value |
|---|---|
| Learning rate | 0.01 |
| Epochs | 1000 |
| Discount factor ($\gamma$) | 0.9 |
| Convergence threshold ($\epsilon$) | 0.01 |
| Weight initialization ($w$) | Normal |
| Optimizer | Adam |
| Neural network structure | (12, 3, 3) |

Table 3: Hyperparameters for Deep-IRL.

### C.2 Descriptiveness of User Policy

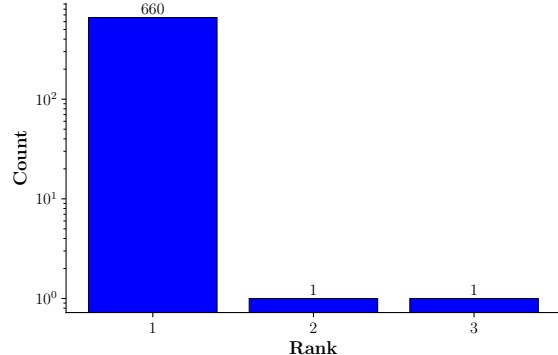

Figure 12: Validation of user policies against 1,000 randomly generated policies.

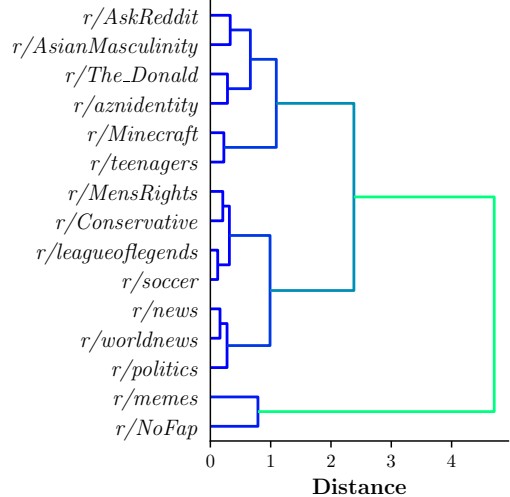

**Figure 13: Dendogram of SWKL.**

User policies do not account for topic content, which can lead to state and action spaces that are either too sparse or too dense, making it difficult to generate unique, distinguishable policies. We validate the descriptive accuracy of inferred policies by comparing the log-likelihood of a user's actual trajectory under their own policy against that of randomly generated policies. These random policies are created by sampling a probability distribution over the six possible actions for each of the 12 states. Since the agent's behavior follows the Markov property, the normalized log-likelihood is calculated as a sum over all actions in the trajectory, conditioned on the state. We define the normalized log-likelihood of observing

trajectory $\tau_u = \{(s_1, a_1), (s_2, a_2), \ldots, (s_{|\tau_u|}, a_{|\tau_u|})\}$ under the user policy $\pi$ as

$$\mathcal{L}(\tau|\pi) = \frac{1}{T} \sum_{k=1}^{|\tau_u|} \log \left( \pi(a_k|s_k) \right). \tag{1}$$

This allows us to rank the most likely policy for each user's trajectory by comparing the log-likelihood of each policy on the user's trajectory to all other policies. Intuitively, if the inferred policies accurately describe the user, the user's demonstrated trajectory should be more likely under their own policy than under random policies, meaning their policy should rank near the top.

We test each user's policy against 1,000 randomly generated policies, with the results shown in Fig. 12. The majority of users rank first, indicating that the inferred policies contain descriptive information about the user.

## C.3 Subreddit Clustering with SWKL

In Fig. 3b, we presented the symmetric weighted Kullback-Leibler (KL) divergence across subreddits, showing that, compared to topic homophily, this measure of behavioral homophily reveals greater similarity between subreddits, suggesting similar user behavior across different communities. To further investigate behavioral homophily, we performed hierarchical clustering, with the resulting dendrogram shown in Fig. 13.

The groupings reinforce several key findings, such as the connections between current events and political subreddits–*r/politics*, *r/news*, and *r/worldnews*–which feature significant numbers of "Disagreers." We also observe behavioral similarities between users of *r/soccer* and *r/leagueoflegends*. Additionally, *r/NoFap* and *r/memes*, two subreddits with weak internal behavioral alignment, form a distinct cluster apart from the others.

