# OpenReview forum: "Behavioral Homophily in Social Media via Inverse Reinforcement Learning: A Reddit Case Study"
_ACM.org/TheWebConf/2025/Conference — WWW 2025 Oral_

### Official Review · Reviewer_6DeC · 2024-11-10

**Novelty:** 6
**Technical Quality:** 5

**Review:**

The paper aims to investigate behavioral homophily, when users exhibit similar behaviors, and to understand whether topically aligned users show similar patterns of behavior. The analysis is performed on a Reddit dataset. The results are as follows: (i) five behavioral user types are identified; (ii) behavioral homophily can be quantified; (iii) behavioral homophily generally aligns with topic homophily (positive correlation), but they can diverge.

The methodology discussed in the paper is clear, so the results are reproducible. However, certain steps in the analysis are not well motivated (see the Questions section).

**Questions:**

1. IRL is implemented to infer a user's reward function based on observed behavior. The accuracy of the inferred function is not discussed (could be compared to traditional RL with a benchmark example).

2. The paper presents the results based on the analysis of only 662 users of ``the 6th most visited website in the world''. Isn't this number too small to generalize?

3. When using an MDP to model behavior, the data must be consistent with the property that past behavior does not affect the current action, only the current state affects. Has this property been verified? If not, modeling with the MDP is not justified.

4. A timestep (p.5, l.532) is not specified.

5. When justifying the SWKL measure, user visited states are weighted, but at the same time both users u and u' have an equal weight of 1/2, yet they can potentially have completely different behavior/state patterns.

6. Why cosine similarity? Any motivation for this choice? How can we ensure that the results are robust to other similarity measures? The dataset covers seven years, so users' preferences for topics may change, certain topics may no longer receive users' attention.

7. The choice of k = 5 for the number of clusters is suggested as a tradeoff between the Silhouette score and the Gap statistic, but the procedure for choosing this value is somewhat subjective, yet transparent.

8. The interpretation of Figure 5c should be confirmed by statistical tests (second paragraph of Section 4.4) and not summarized only from visual aspects.

**Reviewer Confidence:**

3: The reviewer is confident but not certain that the evaluation is correct

**Scope:**

4: The work is relevant to the Web and to the track, and is of broad interest to the community

---

### Official Review · Reviewer_31W7 · 2024-11-23

**Novelty:** 5
**Technical Quality:** 6

**Review:**

This paper introduces a method for measuring homophily in online social networks, offering an alternative approach for situations where building interaction networks is difficult and where topical homophily is less useful on platforms like Reddit, which are organized around specific topics. While the approach is interesting, there are some concerns:

1. The practical implications of this measurement are not clear. The authors state:

"Understanding behavioral homophily is crucial for analyzing network cohesion and societal impact. It offers insights into user interactions and content exposure, with implications for improving recommendations, mitigating polarization, fostering inclusive networks, and designing effective interventions."

However, these points are not well explained. For example, it is unclear how behavioral homophily, inferred from the sequence of states visited and actions taken by users, could be used to mitigate polarization. Similarly, how this could foster inclusive networks is not clear.


2. The final dataset includes only 662 users for analysis. The representativeness of this sample is not clear considering they draw the sample from 7 years of data, and the limited size raises questions about the generalizability of the findings.


Additional comment: Figure 2 is well-designed and effectively explains the framework for inferring behavioral homophily.

**Questions:**

1. Could you provide more details on the practical implications of your method for measuring behavioral homophily? You mentioned: "Understanding behavioral homophily is crucial for analyzing network cohesion and societal impact. It offers insights into user interactions and content exposure, with implications for improving recommendations, mitigating polarization, fostering inclusive networks, and designing effective interventions." For example, how could this method help mitigate polarization or foster more inclusive networks on platforms like Reddit?

2. Considering that your final dataset includes only 662 users, could you clarify how representative this sample is of the broader Reddit user base? Is there a possibility of expanding the sample size or addressing any limitations in the representativeness of your findings, especially considering the 7-year data period?

**Reviewer Confidence:**

3: The reviewer is confident but not certain that the evaluation is correct

**Scope:**

4: The work is relevant to the Web and to the track, and is of broad interest to the community

---

### Official Review · Reviewer_833m · 2024-11-28

**Novelty:** 4
**Technical Quality:** 5

**Review:**

This paper presents a framework based on Inverse Reinforcement Learning (IRL) to measure behavioral homophily in social media. It models user interactions as policies inferred from state-action trajectories and applies this methodology to Reddit, a platform that lacks explicit follower relationships. The findings demonstrate that users’ behaviors can align independently of the topics they discuss, offering a behavioral perspective on homophily that emphasizes actions over content. While applying IRL to homophily provides a fresh perspective, the contribution appears incremental, building on existing methods. Additionally, concerns about the framework's practicality and scalability in real-world settings remain.

Pros:
- Good Application of IRL: The paper proposes a framework that leverages Inverse Reinforcement Learning to evaluate behavioral homophily in social media. This approach provides a new way to quantify homophily based on user behavior rather than content. Applying IRL to Reddit, the authors show that users’ actions are often aligned even when discussing different topics, highlighting a form of homophily that goes beyond topical similarity. This contributes to the understanding of how users interact within online communities without explicit social ties.
- Practical Insights: The case study on Reddit offers practical insights into user behaviors across different communities, with the analysis showing that homophily manifests differently across topical and behavioral dimensions. This offers valuable guidance for future research into social media dynamics and how homophily contributes to user interactions.
- Good Presentation: The paper is well-structured, logically organized, and easy to follow.

Cons:
- Dependence on Data Labeling: The method heavily relies on Data Labeling steps, particularly for argumentation classification. This raises concerns about its feasibility in real-world applications where effective classifiers may not be readily available. The quality of these classifiers directly impacts the quality of inferred user policies. Given this reliance, it would be useful to analyze whether the method produces stable results for behavioral homophily when the accuracy of the classifiers in the data labeling step fluctuates, to assess the robustness of the proposed approach in more variable settings.
- Running time analysis: The computational cost of applying IRL to large datasets may limit the practicality of the proposed method for broader applications.  It is recommended to report the training and inference time for the proposed method and its baseline comparisons to provide a clearer picture of the trade-off between accuracy and computational cost.
- Platform-Specific Focus: The study is focused exclusively on Reddit, which raises concerns about the generalizability of the findings to other platforms with different structures and interaction paradigms.

**Questions:**

- Could you provide a running time and memory analysis of the method to better understand its computational requirements and scalability?
- Will the method produce stable results for behavioral homophily when the accuracy of the classifiers in the data labeling step fluctuates?
- How might the findings vary when applied to different social media platforms with distinct structures and interaction paradigms?

**Reviewer Confidence:**

2: The reviewer is willing to defend the evaluation, but it is likely that the reviewer did not understand parts of the paper

**Scope:**

4: The work is relevant to the Web and to the track, and is of broad interest to the community

---

### Official Review · Reviewer_8JzY · 2024-11-29

**Novelty:** 6
**Technical Quality:** 6

**Review:**

The paper focuses on the behaviour of Reddit users, and explores to what extent such behaviour is correlated with the subreddit the users belong to. To do so, authors introduce the concept of behaviour homophily (i.e., how much users are similar based on their actions on the platform) based on Inverse Reinforcement Learning (IRL), and discuss it together and in comparison with topic homophily (i.e., how much users are similar based on which topics they discuss). Results show that behavioural homophily is able to capture users' behaviour in a way that is inaccessible to topic homophily.

I believe the introduction of behavioural homophily offers a new, valuable tool to researchers, that could be useful also outside the Reddit use case. Its utility is confirmed by the results offered by the authors. Below, you can find some of my notes on the paper.

Some sentences should be substantiated by referring to previous works (l. 67, l. 70)

A quick Scholar query returned other works that could potentially be of interest as related works, intersecting homophily and Reddit. They include, but are not limited to:
- https://doi.org/10.2196/41882
- https://doi.org/10.1177/0165551520979869
- https://doi.org/10.1177/0093650218813655

I am not suggesting that these works must necessarily be cited (and please be reassured that the reviewer is not involved in any of them), but I believe some of them could help complete you Related Work section.

I believe that Section 1.1 and Section 1.2 could be reunited in one, as challenges and contributions are strictly related.

In ll. 138-140 authors write that "[IRL] application to social media has been more limited due to challenges in encoding". Are nonetheless works that use it? Can authors point at them? It has been limited, or it does not exist, to the best of authors' knowledge?

At the end of Section 2.2, authors mention that the training parameters are detailed in the Appendix.

In ll. 319-320, authors state that "While a user’s overall karma is displayed on their profile, it does not affect functionality". However, as far as I know, some communities require minimal karma threshold to perform actions, e.g., post in the subreddit. I would rephrase the sentence in that sense.

In l. 346, authors present heterophily as anti-homophily, but they have already mentioned heterophily in l. 165. I would move the definition in the latter, or repeat it.

The choice of the seed subreddits seem a little arbitrary. Authors should discuss more clearly the reasons behind their choice.

In Section 3.2, authors mention an "activity level", used to rank users. However, it is not clear how this activity level is computed.

Section 3.2 is the most obscure. The definition of "initial setting" is misleading to me, as the set is not initial, but the result of sampling from the yearly rank lists. Also, the numerical aspect is not clear: as far as I have understood, sampling 250 users from 13 seed subreddits for 7 years create ~18,000 unique users (22.750 total users, some of them duplicated), who are then again sampled with a 1/3 rate to obtain the intermediary set. I suggest to refactor the section, use different (even non-ambiguous, symbolic) names for the sets, and explain in clearer terms the numerical pipeline from yearly ranks to the final user set.

I have troubled understanding how you calculated Subreddit Topic Homophily. Authors first ask how "activities across Reddit align in terms of topics" for "users [who] primarily engage with the same subreddit" (ll. 619-620). Then, in l. 623, authors consider "home users two distinct subreddits" (aside, I think there is a typo or a word missing in the sentence), confirmed by the fact that the two sets $C \cap C' = \emptyset$. I suggest to rephrase the section to better express the intentions.

In ll. 663-664, authors include r/soccer and r/leagueoflegends among "topics such as politics and activism". I would rephrase, or rather explain the presence of these two subreddits, which seem to fit better among the "gaming and youth topics" (l.665).

In Figure 3c, the scale is not indicated. It may also be useful to rescale the length of the radius, to better appreciate the difference towards the centre of the plot.

In Figure 4, I encourage the use of hatches, together with colours, to distinguish bars: it would help b/w and colour-blind readers.

Ll. 744-747 could be substantiated with some numbers (e.g., % of agreeing comments).

It is not clear if the "topic vector" introduced in l. 800 and the "home subreddit topic vector" (l. 807) are the same object. Please clarify, or, even better, introduce a clear definition of these quantities.

Figure 5b is not useful at all, since points overlap with each other and colours are too similar one to the others. Even if patterns existed, it would be very difficult to spot them. Authors should evaluate another visualisation, e.g., boxplots or barplots? Or even simply write in the text, as the figure does not add much.

In Figure 11, I don't understand the role of the green, dashed line with label "Optimal k: 10".

Minor comments:

- Be consistent with hyphenation: authors sometimes use two-dashes hyphens between words, sometimes three-dashes ones. I suggest using three dashes (https://www-users.york.ac.uk/~pjh503/LaTeX/misc.html). Somewhere authors leave spaces around hyphens, sometimes they do not: again, I suggest consistency.
- "who connects" not clear in ll. 195-196.
- extra space not needed when writing 6,000 in l. 390.
- extra space needed after "." in l. 912.
- extra dot needed after ending quote in l. 620 and l. 782.

**Questions:**

# Seed subreddits

Why did you precisely choose the seed subreddits? Are there past works exploring multiple subreddits with which you can compare your approach?

# Activity level

How is the activity level of a subreddit computed?

# Unavailable content

What's the (relative) volume of the unavailable content you mention in ll. 447-450?

# Root comment stance

Is yours a deliberate choice not to split IRC and RC into different agreement stances, such as $IRC_{+}$, $IRC_{-}$, $IRC_{\sim}$ and $RC_{+}$, $RC_{-}$, $RC_{\sim}$, as you did with IR and PR? An agreement stance could be estimated with respect to the initial post in the thread.

**Reviewer Confidence:**

3: The reviewer is confident but not certain that the evaluation is correct

**Scope:**

4: The work is relevant to the Web and to the track, and is of broad interest to the community